# Enhancing the knowledge of parents on child health using eLearning in a government school in the semi-rural community of Karachi, Pakistan

**Saleema Gulzar**[1]*, **Sana Saeed**[2], **Salimah Taufiq Kirmani**[3], **Rozina Karmaliani**[4]

**1** School of Nursing and Midwifery, The Aga Khan University, Karachi, Pakistan, **2** Department of Pediatrics and Child Health, The Aga Khan University, Karachi, Pakistan, **3** Department of Information Technology, The Aga Khan University, Karachi, Pakistan, **4** Department of Community Health Services, School of Nursing and Midwifery, The Aga Khan University, Karachi, Pakistan

* saleema.gulzar@aku.edu

**Data Availability Statement:** Contact details for ethics committee where data requests may be sent: erc.pakistan@aku.edu.

## Abstract

Education is one of the vital social determinants of health. Health and education share a symbiotic relationship for all cadre including children and adolescents to ensure that they are well equipped to combat the health risk in the environment. The current literature globally found some initiatives to create health awareness among school children. However, there is a dearth of studies available addressing parental health awareness through school platforms. Therefore, the current study aims to fill this gap, and the Aga Khan University School of Nursing and Midwifery initiated the School Health Program (SHP) in one of the remote communities in Sindh, Pakistan. The overall goal of the study was to improve children's health by enhancing the health awareness of the parents through school platforms utilizing online modalities. Another objective of this study was to identify the effect of using eLearning on parental knowledge and perceptions. The study utilized a sequential explanatory mixed-method design. Twelve health awareness sessions relevant to children's health using eLearning were conducted over one year. Parents' knowledge was assessed through a pre-posttest, which was administered after each teaching session. Subsequently, focused group discussions were carried out with parents, community leaders, and schoolteachers to gain insights regarding the effectiveness of the health education program. The pre-and post-test results showed again in knowledge in nine out of twelve sessions. The findings from qualitative content analysis yielded three key themes: Perceived usefulness of eLearning, Barriers affecting usability, and Way forward for eLearning through school platforms. The study showed parental satisfaction with the online health education awareness program. They exhibited enthusiasm and desire for further similar sessions in the future. The results demonstrated an enhancement in parental awareness about common health conditions among school children. This study may be replicated on a larger scale in the schools of Pakistan.

**Funding:** The author(s) received no specific funding for this work.

**Competing interests:** The authors have declared that no competing interests exist.

## Introduction

Health education is one of the most critical components of universal health coverage [1]. Studies have shown that parents who are better educated about health care are better equipped to cater to the developmental needs of their children [2]. Students whose parents are actively involved with their health are reported to achieve better academic outcomes [3]. Further, the children are likely to be more participative, attentive, and motivated to perform well in their studies. In addition, a better connection between home and school means that more children are likely to complete their education on time, resulting in lower drop-out rates [4]. On the other hand, parents who are not well informed about the health and developmental needs of their children, are also less likely to be involved in their child's growth and development [5]. Therefore, creating health awareness among parents regarding disease prevention and health promotion of school-aged children would lead to better health outcomes [6, 7]. The available literature has depicted the impact of health education among children and adolescents globally, however, creating health awareness among parents of school children has not been paid much attention in Pakistan. Therefore, the current study was carried out to create awareness among parents of school children because parents influence more on a child's behaviors [8].

Pakistan being a low-resourced country faces challenges in imparting health education among parents due to a lack of educational reforms which include parents at the core. In several peri-urban and rural areas of Pakistan, the child mortality rate is higher than in many Low and Middle-Income Countries (LMICs). According to Masquelier et al. (2018), the mortality rate in Asia and Africa for the age group 5 to 14 is the highest in the region [9]. Similar results are reported in which multiple reasons for child morbidity and mortality were reported, with lack of awareness amongst parents being amongst the most important factors highlighted [10, 11].

School provides a powerful platform to make health accessible for school children. It deals with the curative, preventive, and promotive aspects of health through school settings with the purpose to maintain the health and well-being of children by detecting and treating common illnesses before it gets too complicated. Gulzar et al. (2017), is a success story of a school health program in the local context [12]. The school health curriculum was developed and implemented for students in a higher secondary school in urban Karachi. The study identified a vital gap that has occurred due to the lack of including parents in such programs. There is a paucity of literature regarding initiatives that are directed to improve healthcare awareness among parents and technology has rarely been considered as a possible solution to address this gap, which is the focus of the current study.

One such project was started in a semi-rural community in Sindh. This initiative aimed to facilitate the provision of basic health services to the school children and increase the awareness of the caregivers or parents of school children with the basic health knowledge related to childhood illnesses and care to address the gap in the literature in our local context.

In most cases, parents are responsible for children's primary years of education [13]. Several studies have proven internationally that effective primary care and healthy communication with parents result in improved attendance and engagement at school [14]. Therefore, this project launched a health education program for parents using an eLearning approach in 2017. The infrastructure for the online health education program was developed in the selected school setting. The health education sessions were carried out by content experts relevant to child's health. The online session was conducted once every month to educate parents on common health conditions of children.

This study aims to evaluate the effect of eLearning-based school health education programs provided in the school in a semi-rural community setting to enhance parental awareness to promote the health of school children.

## Materials and methods

The study was conducted in one of the government schools (grade Early Child Development (ECD)-grade VIII) of the semi-rural community in Sindh, Pakistan, in the year 2016–2019. This study utilized the mixed-method sequential design. To sensitize and mobilize the community, the team of healthcare providers worked alongside the community leaders and school administration in the community. To get the baseline assessment, initially, a formal meeting was carried out with parents and teachers of grades ECD-, i.e., primary, and secondary grades, to identify the common health problems of children that parents found challenging to deal with. The consensus of the parents regarding the time and venue of the health education sessions was sought through an in-person meeting with parents during the need assessment meeting. The total number of students in the school was 253 (ECD-grade VIII). For the current study, parents were invited and asked to voluntary participation in a health awareness program, generally, each parent has three to six children studying at the same school. All the parents who showed up on the day of the session were included through the universal sampling method. Based on this baseline data, the healthcare professionals (nurses, physicians, and nutritionists.) who had the content expertise were approached at one of the private university hospitals in Karachi to facilitate the live session for the parents. The sessions were scheduled once a month on the parents' preferred day and time. The parents were informed and sent reminders prior to the sessions, and they gathered in the school to attend virtual sessions using MDConsults software. The software helped us cater to the issue of accessibility by parents and travel issues of the session facilitators. The technological solution has assisted the target community in two ways. Firstly, it has helped to mitigate the issue of availability and accessibility of internet and electronic devices individual homes to join in the virtual learning program. Secondly, it has provided the feasibility to the community to attend the sessions virtually in their community that has saved their travel cost and time. Each session was interactive and had pictorial representations for better understanding. All parents who consented to the study were included in the study.

## Ethics approval and consent to participate

This study is approved by The Aga Khan University Ethical Review Committee (4410-SON-ERC-16). All individuals were informed of the ethical issues and were given the opportunity to withdraw from the study at any time. Written informed consent was obtained from all participating individuals.

## Data collection

**Quantitative data collection and analysis.**   In the quantitative arm, the knowledge of the parents was assessed through pre and post-tests. A specific questionnaire was developed for each health education session, in the local language (Urdu) such as immunization, and the importance of a balanced diet for children. The multiple-choice questions were developed with the help of a content expert who facilitated the sessions. Each questionnaire comprised of 8–10 items with four options. The caregivers/parents had to make the appropriate choice on each question S1 Text. The selected school has one onboard female school health nurse, responsible for administering the pre-posttest questionnaire. In cases of queries, she addressed the participants' concerns. The data was entered and analyzed using SPSS 20.0 software. Paired t-test was applied to measure the effect of the intervention on the pre-post scores. P-value of <0.05 was taken as significant.

**Qualitative data collection and analysis.**   In the qualitative arm of this study, the data were gathered through our four focus group discussions (FGDs). These comprise two FGDs

with parents and one each with community leaders and teachers separately, were conducted by the research team members, who possessed the rich experience and relevant qualifications for conducting FGDs. The FGDs were conducted and audio-recorded after completing 12 online sessions to explore in-depth perspectives about these sessions. The parents, teachers, and community leaders were invited to share their experiences related to the online health education program. The data was transcribed and translated from Urdu to English. The data analysis was carried out by the research team members who were master's and PhD prepared nurses with a research background in both quantitative and qualitative research approaches. The qualitative data analysis was carried out independently by two expert researchers, adding trustworthiness to the findings. Similar codes were combined to generate themes, within which subthemes were extracted.

# Results

## Quantitative pre-post assessment results

The quantitative arm depicted that the mean participation of parents in the eLearning sessions was found to be 21.8 (max: 34 & min: 13: SD 7.2). There was a significant difference in parents' knowledge after eight sessions (N = 12, 66%) as measured through a paired t-test. However, no statistical difference was found in pre-post knowledge assessment on the topics of child safety, nutrition, and substance abuse as shown in Table 1. The set of questions from each session was developed by the content experts and reviewed by the team for contextual reference. The translation of the tool into the local language by a team member who possessed bilingual language skills. The questionnaire was multiple-choice, and each participant was given one mark for the correct answer. In the sessions, the number of participants ranged from 15–34 per session. In 8 out of 12 sessions including vaccination, diarrhea, puberty, pneumonia, seizures, personal hygiene, measles, and cognitive development depicted a significant increase in the knowledge of participants.

## Qualitative data results

Content analysis of the responses revealed three key themes that are depicted in Table 2.

**Table 1. Pre & post analysis of e-learning sessions for mothers.**

| E-learning Modules | Number of Participants | Mean Difference (SD) | P-value |
|---|---|---|---|
| Child Safety | 25 | -0.04(1.20) | 0.870 |
| Vaccination | 22 | 1.59(1.18) | <0.001* |
| Diarrhea | 18 | 0.38(0.77) | 0.049* |
| Puberty | 34 | 0.58(1.30) | 0.013* |
| Nutrition | 30 | 0.13(1.65) | 0.662 |
| Pneumonia | 15 | 0.40(0.63) | 0.028* |
| Seizures | 28 | 1.17(1.27) | <0.001* |
| Personal Hygiene | 13 | 1.38(1.38) | 0.004* |
| Substance Abuse | 15 | 0.60(1.91) | 0.246 |
| Measles | 15 | 2.60(1.35) | <0.001* |
| Parenting | 17 | -0.76(1.20) | 0.018* |
| Cognitive Development | 29 | 0.89(1.51) | 0.004* |

*P-value of ≤ 0.05.

**Table 2. Themes and Subthemes of the study.**

| | Themes | Subthemes |
|---|---|---|
| 1- | Perceived usefulness of eLearning in the community | a) Awareness of common illnesses and home-based management |
| | | b) Dual service under one roof |
| 2- | Barriers affecting the usability of eLearning in the rural community | a) Cultural issues and community mindset technology-based issues |
| 3- | Way forward for e-learning for distant communities | a) Personal growth |
| | | b) Societal awareness |

**Perceived usefulness of eLearning sessions/program.**   The FGD with the students' mothers revealed the usefulness of these sessions in improving their knowledge about common childhood and adolescent health conditions. They perceived that this was by far the only opportunity available for them where they could interact with a healthcare expert facilitating the session remotely and addressing their queries. They shared their satisfaction and gratitude towards the initiative as it not only made them aware of their children's health and well-being but also helped them develop the confidence to manage minor and common health conditions at home.

**Awareness of common illnesses and home-based management.**   Caregivers were informed about the scarcity of healthcare services in their community and shared that they travel two to three hours to seek a health facility. It becomes close to impossible for them to consult those health services, especially for common illnesses such as diarrhea. They perceived that these sessions empowered them not only with the knowledge but also equipped them with certain skills that can be taken at home during illness.

One of the participants shared, "*The doctor/nurse taught us what causes diarrhea, vomiting and how to make ORS (oral rehydration salt), and I used this information at home whenever needed.*"

Another mother shared, "*We didn't know what to teach our daughters about the menstrual cycle. We learned that we don't necessarily require medication during the menstrual cycle, but maintaining hygiene is the most important action. Secondly, if someone is bleeding less, this could be due to blood deficiency.*"

Overall, the participants appreciated this initiative and shared that, generally, these sessions helped them understand the primary causes of the most common health conditions and how to prevent and manage them at early stages.

The mothers also shared that the eLearning sessions helped them and their children to take care of their health and well-being. The transformative experience of their learning about the health outcome of their family has reinforced their trust in this system of learning.

Another mother appreciated by sharing the experience of attending one of the sessions on a balanced diet by saying, "Before this health awareness program, we could not distinguish between right and wrong concerning our children's upbringing, but now our minds have opened up to various ways to maintain our health, what we should eat now, and why combinations of diet are important in our diet, and this information will always stay in our mind."

The participants also shared their experiences about safety measures to be taken for their children. They found these sessions very helpful in taking some preventive measures for the child's safety.

**Dual services under one roof.**   School is one such place whereby the children not only learn new ideas and gain new knowledge, but also these learning spaces helped them to understand the principles of healthy living, their rights, and duties as responsible citizens. The

project utilized the school space as a platform to gather the caregivers/ parents to help them understand an essential aspect of their children's health. This added an emotional aspect to this intervention and broaden their minds to reflect upon the importance of the connection between health and education. The parents shared their satisfaction with this.

One of the parents shared, "it gives us immense satisfaction by seeing our children learning in an environment which cares about them and their needs"

Another parent reported, "it is very difficult for us to commute due to the non-availability of transport services in our area. Had this intervention taken place at some other space, it would have been close to impossible for us to reach there."

## Barriers affecting the usability

Despite poor literacy rates, the caregivers understood and valued this intervention. They expressed their desire of having a longing experience whereby they could learn and improve the health and hygiene status of their families. In the next phase of FGD, participants shared their opinions regarding areas which are needed to be improved for the future in such initiatives.

**Cultural issues and community mindset.**   The participants drew attention to a vital point about maintaining confidentiality related to certain sensitive topics. They informed that discussing those aspects is considered Taboo in their culture. For example, puberty in children is one of the areas which is considered private in their community. They suggested having discussions individually rather than openly in public on those aspects. They shared that discussing those issues in public might have some cultural implications as discussing those matters openly is not considered decent in their community.

One of the schoolteachers informed, "*A session in which the doctor discussed personal hygiene during menstruation, though it was very informative for the mothers, the parents shared their concerns of privacy.*"

**Technology-based challenges.**   Technology plays a vital role in connecting remote areas. The experience of electronic learning is directly proportional to the unhampered connectivity. To make sure that there is no break in the process, the project team arranged all the possible devices and provided the optimum setup. However, sometimes, during the session, there were challenges related to network connectivity during the sessions.

One of the parents shared, "*the whole discussion was going on so well, suddenly the connection dropped, and the doctor disappeared from our screen. Upon rejoining it took a bit of time for me to recall the previous point*".

## Way forward for learning to spread health awareness in rural communities

The participants were found to be enthusiastic and proposed ideas for subsequent sessions. Based on their experience, they identified the need of including various other areas in future activities.

**Societal awareness.**   Being a close community, the individuals utilized some of their time to discuss their challenges, the politics, school performances of their children, etc. The community leaders informed that they had often seen parents discussing those aspects, which are taught to them, among themselves, and hence they felt that this raised the community's awareness and brought about a positive change in knowledge, attitude, and practices.

A participant shared: "*There needs to be a session on family planning as we do not have enough knowledge regarding birth spacing.*"

**Personal growth.**   During the project, it was observed that all stakeholders were very excited and contributed their bit to implement and execute it. One of the most fascinating

observations was that they were not only curious to know about physical health, but they also showed interest in getting awareness about mental well-being. During the discussion, it was evident that the community stakeholders wished to learn about basic lifesaving skills, first aid practices, etc.

Other participants shared: "There should be training about the parent-child relationship and first aid in case of injury."

## Discussion

This study aimed to evaluate the effectiveness of interactive health education programs using eLearning modalities for creating awareness among parents and promoting health and well-being among school-aged children in a semi-rural community of Pakistan.

Our study showed that most of the health education sessions were beneficial and informative for parents, and they were very appreciative of the health awareness program initiated through the school platform. eLearning and health education have been widely studied in various countries and cultures, and their effects have been proven to have a significant impact on parental education, awareness, and health-seeking behaviors [15–19]. Akhter et al. (2015), reported the odds of under-five mortality 8% lower for the children with mothers having secondary education, compared to the children with uneducated mothers [8]. Another study conducted in India reported that only 39% of parents had knowledge and awareness of dental care which was addressed through health education. Similar studies carried out in western countries such as Australia and Spain on health education among parents for their child's health showed significant results in terms of change in parental health practices. This sort of initiative has enhanced self-efficacy among parents to provide better health care to their children [20, 21].

Providing access to health through awareness using eLearning modalities has proved to be a paradigm shift and an important step towards addressing this gap among the parents and community leaders. The resource substitution theory states that health education benefits are greater for people in low-resourced settings [22]. In support of this theory, another study conducted on households in the USA determined that low health literacy affects parent acquisition of knowledge, attitudes, and behaviors, affecting child health outcomes across all the disease prevention [23]. Parents having access to online learning in rural areas would be a vital step forward in reducing health disparities. Literature doesn't report any strong evidence of the use of eLearning for enhancing parental awareness in the local context, however, mobile health has been utilized to improve the attitude and healthcare practices among parents and caregivers in Pakistan [24].

Despite the success of the eLearning sessions, the parents still showed a preference for in-person interactions with the nurses and doctors. This finding was supported by another study conducted in the USA comparing an online and in-person educational program for parents demonstrated that the online session was as effective as the face-to-face program in achieving the required health goals for their children [25]. But the main reason for the preference for in-person interaction was the ability of the health care providers to empathize or understand the parent's feelings better, rather than the gain in knowledge. A similar study carried out in Australia intending to understand the role of technology in a person-centered learning environment found that "eLearning teaching modalities is the key to revolutionizing education in the future," and utilizing technology will open ways to promote literacy [26]. Another study conducted on adopted children in China concluded that maternal education has a positive and causal effect on the child's health, regardless after adding control for income, and other socio-economic variables [27]. A systematic review and meta-analysis further reinforced the

importance and impact of parental education on child development and health. In this study, the results showed that parental education was directly associated with reducing under-five mortality, with the mothers' education being a more robust predictor [11]. Furthermore, a study in China proved that along with health, children with parents with a college education also objectively perform better in academics, owing to the positive interaction between children and their parents [28].

Involving the community leaders and stakeholders was one of the strengths of our study, which helped create a more comprehensive and lasting impact. A similar intervention was implemented in another study where local leaders were involved in encouraging and educating the community, which yielded positive results in the knowledge and decision-making skills of the participants [15].

Along with the overall benefits, this online education program through the school platform has proven that a program like this would help address health care needs by creating awareness among parents where accessibility of health is a significant issue. Using a technology-based health education initiative is a viable solution for providing health education because it is recognized as convenient, efficient, and comparatively economical for the community at large [29]. However, a combination of online and face-to-face may be more effective than exclusively online learning modality alone to improve learning outcomes [30]. Social Reforms are required at the grass-root level (townships, villages, etc.) to educate the community and ensure that the needs of young children in Pakistan are met in this new era of modern technology.

## Limitations

This study is among the few reported evidence from low-middle-income countries such as Pakistan. Though this study has reported the immediate effect of the intervention on parental awareness, our study did not show the subsequent impact on future practices of parents, which is one of the key limitations and an area of future research. Future studies on this subject matter should consider having schoolteachers on the research team utilizing an implementation science approach.

## Conclusion

Overall, the results from the quantitative arm showed significance in terms of increasing the parents' knowledge level. The qualitative arm showed parents' satisfaction with the online awareness program and revealed that it enhanced their awareness level about preventable health conditions of children. The participants showed enthusiasm and desire for more similar sessions in the future. The findings from this study have the potential of scaling it up to include various education setups as well as other communities and school teachers, which will be the focus of our future research.

## Supporting information

**S1 Text. Sample questionnaire.**
(DOC)

## Acknowledgments

The authors expressed sincere gratitude towards Dr. Areeba Hussain, MD for assisting in manuscript formatting and submission.

## Author Contributions

**Conceptualization:** Saleema Gulzar, Sana Saeed, Rozina Karmaliani.

**Data curation:** Saleema Gulzar, Salimah Taufiq Kirmani.

**Formal analysis:** Saleema Gulzar, Sana Saeed.

**Investigation:** Saleema Gulzar.

**Methodology:** Saleema Gulzar, Sana Saeed.

**Project administration:** Saleema Gulzar.

**Resources:** Saleema Gulzar.

**Software:** Saleema Gulzar.

**Supervision:** Saleema Gulzar.

**Visualization:** Saleema Gulzar.

**Writing – original draft:** Saleema Gulzar, Sana Saeed, Salimah Taufiq Kirmani.

**Writing – review & editing:** Saleema Gulzar, Sana Saeed, Rozina Karmaliani.

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
