## [Decision Letter · Decision Letter 0]

27 Jan 2022

PGPH-D-21-00889

Empowering parents of school children using eLearning in a government school of semi-rural community of Karachi, Pakistan

Dear Dr. Gulzar,

Thank you for submitting your manuscript to PLOS Global Public Health. After careful consideration, we feel that it has merit but does not fully meet PLOS Global Public Health’s publication criteria as it currently stands. Therefore, we invite you to submit a revised version of the manuscript that addresses the points raised during the review process.

We look forward to receiving your revised manuscript.

Kind regards,

Md Anwarul Azim Majumder, PhD

Academic Editor

Journal Requirements:

1. Please include additional information regarding the survey or questionnaire used in the study and ensure that you have provided sufficient details that others could replicate the analyses. For instance, if you developed a questionnaire as part of this study and it is not under a copyright more restrictive than CC-BY, please include a copy, in both the original language and English, as Supporting Information.

2.  If you have no competing interests to declare, please state "The authors have declared that no competing interests exist".

Reviewers' comments:

Reviewer's Responses to Questions

**Comments to the Author**

1. Does this manuscript meet PLOS Global Public Health’s publication criteria? Is the manuscript technically sound, and do the data support the conclusions? The manuscript must describe methodologically and ethically rigorous research with conclusions that are appropriately drawn based on the data presented.

Reviewer #1: Yes

Reviewer #2: Partly

2. Has the statistical analysis been performed appropriately and rigorously?

Reviewer #1: Yes

Reviewer #2: No

3. Have the authors made all data underlying the findings in their manuscript fully available (please refer to the Data Availability Statement at the start of the manuscript PDF file)?

Reviewer #1: No

Reviewer #2: No

4. Is the manuscript presented in an intelligible fashion and written in standard English?

Reviewer #1: Yes

Reviewer #2: Yes

5. Review Comments to the Author

Reviewer #1: 1. Title:

a. Does not reflect what has been discussed in the introduction i.e., health education.

b. Empowering parents of …. What???

2. Abstract:

a. Introductory sentence should highlight the importance of school health programme rather than PHC and health education in general.

b. Rewrite the keywords according to the MeSH terms.

3. Introduction:

a. Poorly written, lack of flow and key topics were not discussed.

b. For example, impact of eLearning-based school health education programs was not discussed. Evidence from regional countries should be documented.

c. Common health problems and issues of children in different countries were not adequately discussed.

d. The need for health education and the present status of health education in primary and secondary schools in Pakistan/regional countries require further discussion.

4. Methods:

a. Is this a primary or secondary school?

b. Who developed and finalized the questionnaire?

c. Who conducted the FGDs? Are they trained and/or expert in conducting FGDs?

d. Were the researchers trained/experts to conduct data analysis?

5. Results:

a. Sub-themes (Table 2) were not discussed clearly.

b. The word “empowering” was mentioned in the title; However, it was not mentioned/discussed either in the introduction, results, or discussion sections.

6. Discussion

a. The key findings need to support by evidence/study findings from Pakistan, South-East Asian and developing countries in addition to developed countries.

7. Conclusion

a. Not included

b. Summarize the key findings and include specific recommendations based on the findings of your research.

Reviewer #2: I think this manuscript is a good concept and will add to the growing body of knowledge on virtual learning. It is also uniquely targeting parents.

I have however noted the following from my point of view:

line 83 - started in a semi-urban to replace started a semi-urban ( minor typo I guess).

line 98- In the methods section, I would have expected to see the population of students stated and the target population of parents calculated. I am of the view teachers should have been included in the initial formal meetings as key stakeholders.

line 99- I think the health care professionals who reviewed the baseline data should be better defined in terms of number and expertise.

line 101 - I am not sure how this worked out. Parents preferred time or agreed time? I would have expected to see the number invited for the sessions to compare with the number participating.

line 104- You may wish to elaborate more about the virtual software and situate it on a specific type of e learning. It may be important to define e learning in the introduction and specify the type used in this study.

lines 111/112 - please verify if the school health nurse was female before using the pronoun 'she', otherwise the statement should be gender neutral.

lines 118-121 - How many FGDs were conducted? what was the composition of each group in terms of homogeneity and heterogeneity (were parents, community leader and teachers mixed or separated) in the groups? How was FGD recorded?

line 211- Please write acronyms in full when used for the first time and provide the acronym in parentheses before subsequent usage.

On a general note, the study did not have examples from similar income country context for comparisons. It may also be helpful to state the availability and accessibility of internet to the parents and target communities. The ownership an access to appropriate electronic devices to join in the virtual learning are also important consideration.

Future studies on this subject matter should consider having school teachers on the research team and considering utilizing an implementation science approach

6. PLOS authors have the option to publish the peer review history of their article (what does this mean?). If published, this will include your full peer review and any attached files.

**Do you want your identity to be public for this peer review?** For information about this choice, including consent withdrawal, please see our Privacy Policy.

Reviewer #1: **Yes: **Dr Sayeeda Rahman

Reviewer #2: No

---

## [Editor Report · Decision Letter 1]

29 Apr 2022

Enhancing the knowledge of parents on child health using eLearning in a government school of the semi-rural community of Karachi, Pakistan

PGPH-D-21-00889R1

Dear Dr. Gulzar,

We are pleased to inform you that your manuscript 'Enhancing the knowledge of parents on child health using eLearning in a government school of the semi-rural community of Karachi, Pakistan' has been provisionally accepted for publication in PLOS Global Public Health.

Best regards,

Md Anwarul Azim Majumder, PhD

Academic Editor